# A Facile and Sustainable Enhancement of Anti-Oxidation Stability of Nafion Membrane

**DOI:** 10.3390/membranes12050521

**Published:** 2022-05-13

**Authors:** Prem P. Sharma, Dukjoon Kim

**Affiliations:** 1School of Chemical Engineering, Sungkyunkwan University, Suwon 16419, Gyeonggi, Korea; 2School of Chemistry, Faculty of Basic Sciences, Shoolini University, Solan 173229, Himachal Pradesh, India; premsharma15@gmail.com

**Keywords:** radical scavenger, Nafion, proton conductivity, anti-oxidation stability

## Abstract

^•^OH radicals are the main cause of chemical degradation of Nafion membranes in fuel cell operation. Although the cerium ion (Ce^3+/4+^, Ce) is reported as an effective ^•^OH radical quencher, its membrane application has critical limitations associated with the reduction of membrane proton conductivity and its leaking. In this study, the Ce-grafted graphitic carbon nitrides (g-C_3_N_4_) (CNCe) nano-particles are synthesized and embedded in Nafion membranes to prolong the ^•^OH radical scavenging effect. The synthesis of CNCe nano-particles is evaluated by X-ray diffraction, energy dispersive X-ray analysis, and transmission electron microscopy. Compared with the pristine and Ce-blended Nafion membranes, the CNCe imbedded ones show tremendous improvement in long-term anti-oxidation stability. While the fluoride emission rates of Nafion are 0.0062 mg·cm^−2^·h^−1^ at the anode and 0.0034 mg·cm^−2^·h^−1^ at the cathode, those of Nafion/CNCe membranes are 0.0037 mg·cm^−2^·h^−1^ at the anode and 0.0023 mg·cm^−2^·h^−1^ at the cathode. The single cell test for Nafion/CNCe membranes at 80 °C and 50% relative humidity illustrates much better durability than those for Nafion and Nafion/Ce, indicating its superior scavenging effect on ^•^OH radicals.

## 1. Introduction

In consideration of high power density, zero carbon emission, and relatively low temperature operation, the polymer electrolyte membrane fuel cell (PEMFC) has earned considerable attention in power generation [1,2,3,4]. Nafion is one of the well-known perfluorosulfonic acid-based membranes for PEMFC, as it has high proton conductivity with good chemical and mechanical stability. Because of its low price competitiveness, great efforts have been made to develop hydrocarbon-based membranes as its substituents [5,6,7]. Development of hydrocarbon-based membranes, however, is still lagging because of its limited proton conductivity and high swelling caused by relatively poor hydrophobic–hydrophilic phase separation compared with perfluorosulfonic acid-based ones. In addition, the long-term chemical stability of hydrocarbon-based membranes such as SPEEK-based is not good, and is even worse than that of Nafion membranes [8]. Almost all polymer electrolyte membranes including both hydrocarbon and perfluoro sulfonic acid systems have suffered from chemical degradation by the attack of ^•^OH and ^•^H radicals generated from the reactions between transition metals and H_2_O_2_ [9]. When the chemical degradation of Nafion occurs by those radicals, HF is usually produced from the tertiary carbon of the -C-F bond while the –COOH, –C-S and –C-O-C– groups are de-bonded via complex mechanisms. Chemical degradation of the hydrophilic pendant site as well as the hydrophobic polymer backbone results in a disruption of ionic channel structure of the membranes, leading to the deterioration of membrane properties and cell performance [10,11,12].

In order to prevent the degradation of PEM, incorporation of the hyperactive radical scavengers such as manganese (Mn) and cerium (Ce) has been studied. Cerium, Ce, is a rare earth metal possessing regenerative activity by the presence of the multivalent oxidation states and oxygen vacancies in its lattice site [13,14]. The exceptional reversibility between the Ce^3+^ and Ce^4+^ states helps to scavenge the hydroxyl radicals (^•^OH) [15,16,17]. While Ce^3+^ reacts with ^•^OH to generate Ce^4+^ and water, Ce^4+^ undergoes a reduction reaction with ^•^OOH and H_2_O_2_ to produce Ce^3+^. The long-term anti-oxidant stability of PEM can be expected from such reversible reactions. Among all scavengers, Ce is found to be most effective, bearing almost ten times higher scavenging strength and twice higher neutralization tendency toward ^•^OH than Mn [18,19]. However, the simple addition of Ce into an ionomer may cause a couple of problems during the fuel cell operations as follows: (i) Ce ions are easily mitigated from the membrane because of its water-soluble tendency; (ii) the positive charge in its oxidative state inhibits the activity of sulfonic acid groups of the ionomer. Hence, recent research in the development of radical scavengers is focused on avoiding (i) the mitigation of Ce from the membrane and (ii) the direct interaction of Ce with the sulfonic acid groups of membranes.

Graphitic carbon nitride (g-C_3_N_4_, CN) has been widely used for energy storage applications because of its redox catalytic activity [20,21]. Its nitride form is stable in the harshest of environments due to its excellency in mechanical, chemical, and thermal stability [21,22]. An abundant number of -N-H functional groups and the lone pair present on the nitrogen atom of the tris-triazine unit provide an extended bonding site as well as Bronsted acid and Lewis base, which helps to retain water for the interconnected channels. Additionally, its large surface area provides an opportunity to accommodate multivalent metal ions. The delocalized π-electrons act as a medium for the ionic interaction with the valance sites of metal elements such that they can be rigidly fixed. For these reasons, graphitic carbon nitride has been chosen here as a host material for Ce such that its mitigation is prohibited to expect a long-term scavenging effect.

This study aims to improve the long-term anti-oxidation stability and cell performance of Nafion by incorporating the hybrid g-C_3_N_4_/Ce (CNCe)-based nano-particles synthesized by a simple calcination of CeO_2_ and melamine. The CNCe nano-particles improve its water retention capability by reducing the direct interaction of Ce with the sulfonic acid groups of Nafion. As the ionic interaction between Ce with π-electrons in g-C_3_N_4_ helps to keep Ce ions intact, it thus avoids their leaching out of the membrane. The effect of CNCe nano-particles on the anti-oxidation stability of the membrane was analyzed by measuring the F^−^ ion emission rate. In addition, its effect on the structure, properties and cell performance of the synthesized membrane was investigated, including the morphology, water uptake, proton conductivity, thermal, mechanical, and dimensional stability.

## 2. Experimental

### 2.1. Materials

Nafion solution (5 wt%, EW 1100) in a mixture of isopropyl alcohol and water was purchased from Dupont (Wilmington, DE, USA). Melamine powder and cerium (III) nitrate hexahydrate were purchased from Sigma Aldrich (St. Louis, MO, USA). Platinum (nominally 40% on carbon black, HiSPEC 4000) was purchased from Alfa Aesar (Ward Hill, MA, USA). 2-Propanol (IPA) and hydrochloric acid (HCl) were purchased from Samchun pure chemical company (Daejung, Korea). Sodium hydroxide pellets (NaOH) were purchased from Sigma-Aldrich (Milwaukee, WI, USA).

### 2.2. Synthesis of Ce Doped g-C_3_N_4_ Hybrid Material

A predetermined amount of cerium(III) nitrate hexahydrate (0.2 g) was dissolved in 10 mL of deionized water. Melamine (1.0 g) was added to the solution for ultra-sonication for 30 min. The reaction mixture was exposed to air at 90°C for 30 min to evaporate water under continuous stirring. The resultant white color solids were crusted to obtain a fine powder and then heated in crucibles at 535 °C in a muffle furnace for 3 h. The final light-yellow product was Ce-doped g-C_3_N_4_ (CNCe), as shown in Figure 1.

### 2.3. Synthesis of CNCe Incorporated Composite Membrane

CNCe (0.3 g) was subjected to sonication in DMSO for 60 min, followed by continuous stirring at 300 rpm for 48 h. The resulting mixture was transferred to the Nafion ionomer solution for further stirring for 24 h. After complete blending, the homogenous mixture was cast on a clean flat petri dish to achieve a thickness of ±80µm. The cast solution was kept in a vacuum oven at 80 °C for 24 h, followed by at 120 °C for 2 h to completely remove solvents. The membranes were peeled off from the petri dish and then dipped into 1 M H_2_SO_4_ solution for ionization of functional groups. The finally synthesized membranes were designated as Nafion/CNCe-x, where x represents the wt% of CNCe. The concentration of CNCe was kept at 6 wt% with respect to the Nafion ionomer, because its concentration beyond 6 wt% caused decrement of proton conductivity due to the destabilization of the sulfonic acid groups.

### 2.4. Characterization

#### 2.4.1. Chemical Structure Analysis

Fourier-transform infrared (FT-IR) spectra were recorded using Perkin-Elmer FT-IR spectroscopy (Nicolet iS10, Brucker IFS 66/S, Brucker, Bremen, Germany) in the wavenumber range of 4000–400 cm^−1^. The shape and size of CN nano-sheets and the morphology of composite membranes were evaluated using field emission scanning electron microscopy (FE-SEM, EM, Phillip XL30 ESEM-FCG, North Billerica, MA, USA). The crystalline structure of both CN and CNCe was determined by X-ray diffraction spectroscopy (D8 Advance, Bruker, Billerica, MA, USA) with CuKα radiation (λ = 1.54 Å). The energy dispersive X-ray spectroscopy and high-resolution transmission electron microscopy (EDS/HR-TEM, JEM-ARF 200F, Tokyo, Japan) were employed for elemental mapping and high-resolution micro-images of composite membranes.

#### 2.4.2. Ion Exchange Capacity (IEC)

The IEC of the membranes was calculated by the acid–base titration method. The membrane samples were washed with DI water and then completely dried to measure their weights (in gram) before immersion in 1.0 M NaCl solution for the complete exchange of H^+^ into Na^+^ ions. The solution was then titrated with 0.1 M of NaOH solution using phenolphthalein as an indicator. The IEC (meq. g^−1^) of membranes was calculated from Equation (1):(1)$$\mathrm{IEC}=\frac{C_{\mathrm{NaOH}}\times V_{\mathrm{NaOH}\;}}{{\mathrm{Dry}}_{w}}$$
where C_NaOH_ and V_NaOH_ are the concentration and volume of titrated NaOH solution and Dry_w_ is the weight of the dry membrane.

#### 2.4.3. Proton Conductivity

Membranes were immersed in water and then cut into 3 cm (length) × 1 cm (width) × ~80 μm (thickness) to measure proton conductivity. The sample was placed in the 4-probe cell (BEKKTECH, Loveland, CO, USA) to measure the in-plane proton conductivity using alternating current (AC) impedance spectroscopy (Zahner IM6e, Germany) with a frequency range from 1 Hz to 1 MHz under 100% relative humidity. The bulk resistance of the membrane was directly obtained from the impedance curve, and the proton conductivity of the membrane was determined from the resistance from Equation (2):(2)$$\sigma =\frac{L}{R\;W\;T}$$

Here, σ is the proton conductivity of the membrane in (S cm^−1^), L is the distance in between the measurement probes in cm, R is the bulk resistance of the membrane in ohm, W is the width of the membrane in cm, and T is the thickness of the membrane in cm.

#### 2.4.4. Water Uptake and Swelling Ratio

Water uptake is an important property because it estimates the hydrophilicity of the membrane. There are two types of water present inside the membrane, which are mainly in the form of bound water and free water. Transportation of ions mostly takes place through the bound water by a hopping mechanism. The membranes were dried at 80 °C for 48 h in a heated oven to obtain its dry weight.

Water uptake was calculated from the following Equation (3)
(3)$$\mathrm{WU}\left(\%\right)=\frac{{\mathrm{Wet}}_{w}-{\mathrm{Dry}}_{w}}{{\mathrm{Dry}}_{w}}$$
where Wet_w_ and Dry_w_ are the weight of the wet and dry membranes, respectively.

The swelling ratio of the membrane was calculated from Equation (4):(4)$$\mathrm{Swelling}\;\mathrm{ratio}=\frac{L_{s}{-L}_{d}}{L_{d}}\;$$
where L_s_ and L_d_ are the length of the wet and dry samples, respectively.

Hydration number, λ, defined by the number of water molecules associated per sulfonic acid group in the proton exchange membrane was also calculated for the water uptake and IEC using the following Equation (5).
(5)$$\lambda =\frac{10\times \mathrm{WU}}{\mathrm{IEC}\times 18.02}$$

#### 2.4.5. Thermal and Mechanical Stability

The stepwise weight loss of the synthesized CN, CNCe, was characterized by thermogravimetric analysis (TGA, Seiko Exstar 6000, Tokyo, Japan). The sample was thermally scanned at a ramping rate of 10 °C min^−1^ from 30 °C to 800 °C under N_2_ and air. The mechanical property of synthesized membranes was analyzed using the universal testing machine (UTM, Model 5565, Lloyd, Fareham, UK) under a load cell of 250 N. The sample dimension was 4 × 1 cm.

#### 2.4.6. Oxidative Stability

Anti-oxidation stability of the synthesized membranes was investigated by measuring the residual weight after Fenton’s test. The completely dry membranes were immersed in Fenton’s solution (3 wt% H_2_O_2_, 4 ppm Fe^2+^) at 80 °C from 24 h. After the samples were taken out of the solution, they were washed several times with DI water and then dried at 80 °C. The residual weight (RW)% was calculated from the difference between the weight of the samples before (m_b_) and after treatment (m_a_) using Equation (6).
(6)$$RW\left(\%\right)=\frac{m_{a}}{m_{b}}\times 100$$

The concentration of fluoride ion (F^−^) in the Fenton’s solution was analyzed using ion chromatography (IC, 882 Compact IC Plus, Metrohm, Switzerland). The fluoride emission rate (FER) was in situ measured to evaluate the chemical stability of MEAs during cell operation. The drain water at both cathode and anode sides of the fuel cell station was periodically collected and analyzed.

FER was calculated according to Equation (7).
FER (µ𝑔𝑐𝑚^−2^ℎ ^−1^) = 𝐶/(𝐴 × 𝑡)(7)
where *C* is the ion concentration (ppm), *A* is the active area of the sample (cm^2^), and *t* is test time (*h*).

#### 2.4.7. Membrane Electrode Assembly and Fuel Cell Performance

The catalyst layer was prepared by mixing 0.1 g of Pt/C (40%), 0.66 g of Nafion ionomer (5 wt% in IPA), 1 mL of DI water, and 8.042 g of isopropanol. The mixture was subjected to a horn-type sonicator (Sonomasher, SL Science, Seoul, Korea) for 30 min for good dispersion. The mixture was sprayed onto a carbon paper by a hand spray pistol to prepare the gas diffusion layer (GDL). The membrane electrodes assembly (MEA) was prepared by pressing the catalyst-coated membrane using a heating press (Ocean Science, Seoul, Korea) at 110 °C and 5 MPa for 3 min. The active area of the MEA for this process was 6.25 cm^2^, and the Pt loading amount for both anode and cathode were 0.5 mg cm^−2^ each. The variation of voltage as a function of current and the voltage drop as a function of time were measured using a unit cell station (SPPSN-300) provided by CNL Energy (Seoul, Korea). During the cell test, hydrogen and oxygen gas was continuously fed to the anode and cathode sites at the flow rate of 0.3 L min^−1^, respectively. The fuel cell performance was measured at 80 °C under 50% relative humidity (RH).

## 3. Results and Discussion

### 3.1. Chemical and Physical Structure of CN and CNCe

The phase structure of CN and CNCe was identified by XRD analysis. In Figure 1a, the occurrence of two dominant peaks at diffraction angles of 13.1° and 27.4° represents the (002) interlayer stacking of conjugated aromatic layers with the hole-to-hole distance of the nitride pores in the tris-triazine structure of CN [23]. The disappearance of those peaks in the CNCe spectrum stems from the intercalation of Ce, which disrupts the crystalline structure of CN during the thermal condensation process. Similar results have been reported from the literature of Ang et al. [24], where the interaction between the TiO_2_ units and the tris-triazine structure caused the interruption of the crystalline structure of melon by decreasing the hydrogen bonding effect [24]. Conversely, the dominant peak at the diffraction angle corresponding to plane (111) represents the overlapping of the characteristics peaks of Ce with CN. Moreover, the appearance of peaks at the Miller indices of (111), (220), and (311) corresponding to the diffraction angles of 28.60°, 33.05°, 48.50° and 59.02° confirms the presence of cubic fluorite structure of Ce.

The structural information of CN and CNCe is further elucidated by FT-IR spectra in Figure 1b. The IR spectrum of CN displays the stretching vibration bands in the region of 800–1700 cm^−1^ [25]. The IR band at 808 cm^−1^ is from the breathing vibration of the tris-triazine ring present in both CN and CNCe. A band at 624 cm^−1^ in the CNCe spectrum confirms the presence of Ce via the vibration of Ce-O stretching [26]. The appearance of the absorption band at 1640 cm^—1^ arises from the vibration of Ce-OH of CNCe, which was absent in CN. All of the above distinguished IR bands of CNCe confirm the incorporation of Ce into CN.

The chemical asserts were also analyzed by the full scan XPS spectra of CNCe in Figure 1c. The reaction with the nitride led to the reduction of cerium, which is confirmed by the presence of both Ce^3+^/Ce^4+^. Figure 1c shows two fitted peaks at 900.30 and 898 eV, assigned to the core levels of Ce^4+^ 3d_3/2_ and Ce^3+^ 3d_3/2_, respectively. Similarly, two more peaks at 888 and 882 eV are assigned to Ce^3+^ 3d_5/2_ and Ce^4+^ 3d_5/2_. It can be observed that the intensity of Ce^4+^ is a little stronger than that of Ce^3+^. The presence of the Ce^3+^ state refers to the oxygen-vacant site in the crystalline form, which is the positive contribution toward anti-oxidation stability by regeneration of the ceria-based hybrid material.

The morphology and elemental composition of CNCe were investigated using HR-TEM. Figure 2a illustrates a regularly stacked morphology of CNCe, which is consistent with other reports [26]. This TEM image shows the spherical shapes of CNCe nano-particles with a nearly uniform diameter of ~30 nm. In Figure 2b, the lattice fringe spacing is 3.12 Å, associated with the (002) and (111) lattice planes of CN and Ce, which indicates the intercalation of Ce into CN. Figure 2c illustrates the EDX results for the mapping of the respective elements where C, N, and Ce are co-existing in good agreement with the XPS results.

### 3.2. Chemical and Physical Structure of Composite Membrane

Figure 3a represents the full scan XPS spectra of the pristine Nafion and Nafion/CNCe membranes. Here, the presence of Ce was assured from the peak at the binding energy of 833.34 eV in the Nafion/CNCe spectrum, which is not shown in the pristine Nafion. The remaining peaks related with O, C and the sp^2^ carbon confirm the well-adhered architecture of the pristine Nafion and Nafion/CNCe. Figure 3b shows the cross-sectional FE-SEM images of Nafion/CNCe membranes with elemental mapping. The membrane surface is quite smooth without any defect. When the membranes are exposed for elemental analysis, the uniform distribution of Ce is illustrated from Nafion/CNCe. This mapping analysis indicates the good dispersion of CNCe in the Nafion matrix without obvious aggregation.

### 3.3. Thermal and Mechanical Properties

To analyze the thermal stability of CN and CN/Ce, thermal gravimetric studies were conducted in the range from 30 to 800 °C at a heating rate of 10 °C min^−1^. As shown in Figure 4a, CN shows excellent thermal stability up to 300 °C, above which its weight slowly decreases due to the oxidation of free nitrogen and carbon present at the peripheral areas of CN. CNCe was thermally more stable than CN, as there was no weight loss up to 600 ^0^C. This is because the van der Walls forces between cerium and heptazine units induces higher combustion temperature.

The effect of the CNCe on the mechanical strength of Nafion is illustrated in Figure 4b. Three different samples with a 1 × 4 cm dimension were tested with UTM under the hydrated state. The elongation at breakage was slightly decreased by introduction of CNCe, as it reduced the water uptake, which has plasticizing effects on the membrane. In the case of Nafion/Ce, the direct inactivation of the sulfonic acid group, due to physical interaction of positive and negative charges present over cerium and the functional group of the polymer, reduces the hydration capacity of the composite membrane, which results in poor elasticity. Conversely, in the case of Nafion/CNCe, the intercalation of cerium resulted in the exfoliation of stacked morphology and thus provided a free space to absorb more water to enhance the elastic property.

### 3.4. Water Uptake and Swelling Ratio of Composite Membranes

The physicochemical properties including IEC, WU and hydration number (λ) of Nafion, Nafion-Ce, and Nafion/CNCe membranes are discussed here. The IEC values decreased from 0.86 to 0.71 meq. g^−1^ by addition of CNCe because of the stacked nature of CN. Moreover, the insertion of Ce is responsible for the further decrement of the IEC value to 0.68 meq. g^−1^ in the case of Nafion/Ce, due to the direct interaction of Ce with the sulfonic acid groups of Nafion.

Similar behavior was illustrated for the hydration number. Because CNCe provides a shielding effect on the sulfonic acid groups of Nafion, the hydration number of Nafion is less decreased by the addition of CNCe than Ce, as shown in Table 1. The water uptake of Nafion was 21.78% at 40 °C, but it was 17.39% for Nafion/CNCe and 12.14% for Nafion/Ce as shown in Figure 5a. This decrement in water uptake is attributed to the inactivation of the diffusion sites in the polymer backbone by the presence of Ce. The swelling ratio followed the same trend as the water uptake, as shown in Figure 5b. The swelling ratio was diminished by the presence of Ce and CNCe because those fillers restrict the volume expansion via water uptake.

### 3.5. Proton Conductivity

Proton conductivity is one of the most important properties of polymer electrolyte membranes for PEMFC application. As shown in Figure 6a, proton conductivity increases with the temperature from 40 to 80 °C because of the thermally enhanced kinetic motion of protons. The proton conductivity was 0.078 S cm^−1^ for Nafion, but it decreased to 0.041 and 0.032 S cm^−1^ for Nafion/CNCe and Nafion/Ce, respectively. The ionic bonding between the acidic groups of Nafion and the positive charges of free amines in CN as well as local agglomeration of the CN nano-particles led to a decreased number of hopping sites. In the case of Nafion/Ce, the positive charges of Ce might be more dominant toward diminishment of proton conductivity, but it was not that affected for Nafion/CNCe, because of the following reasons. The first is the shielding effect provided by CN to the hopping sites of Nafion, and the second is the presence of the π-electron cloud over the tris-triazine ring, which provides a diffusion pathway for proton transport by forming an electron–ion coupling as depicted in Figure 6b. Thus, comparable proton conductivity is achieved by the composite membrane embedded with CNCe nano-particles.

### 3.6. Chemical Stability

Oxidative stability is an important property of the polymer electrolyte membrane, because it is related with the long-term operation of PEMFC. In Table 1, the residual weight of Nafion/CNCe after Fenton’s test for 72 h was 84.39%, which was higher than those of Nafion/Ce and Nafion at 82.55% and 79.62%, respectively. This result shows a great affinity of the Ce ion with CN for prolonged radical scavenging effect. To further evaluate the CNCe effect on the chemical stability of Nafion, the in situ degradation test of membrane electrode assembly is carried out measuring the fluorine emission rate (FER) as a function of time.

Figure 7 represents the time-dependent FER in the OCV test for both anode and cathode sides, as the degradation of fluoropolymer results in the emission of fluorine species from the electrode sites. In the case of the proton exchange membrane, the emission of fluorine is expected to be higher at the anode site due to the faster kinetics of HER in association with the counterattack of ^•^OH radicals. The resulting FER for Nafion was 0.0062 mg·cm^−2^·h^−1^ at the cathode and 0.0034 mg·cm^−2^·h^−1^ at the anode, but that of the Nafion/CNCe membranes was 0.0037 mg·cm^−2^·h^−1^ at the cathode and 0.0023 mg·cm^−2^·h^−1^ at the anode. These results clearly indicate the radical scavenging effect of CNCe on the anti-oxidation stability of membranes. A table of comparison with other Nafion composites has also been added to check the relative performances for the respective membranes.

Moreover, comparison of the performance of the Nafion composite membranes in terms of proton conductivity and cell performances are also listed in Table 2 with the previously reported research works [2,27].

Figure 8 shows the polarization curves measured at the start-up and after 120 h for the MEA fabricated with Nafion and composite membranes. The power density of 233 mW·cm^−2^, which is higher than those of Nafion/Ce and Nafion/CNCe, 130 and 193 mW·cm^−2^, respectively, was obtained for the Nafion membrane at the beginning of the test, as shown in Figure 8a. The presence of nano-particles creates interfacial resistance, decreasing the power density. As the oxidation reaction at the anode side generally occurs faster than the reduction reaction at the cathode side, the reduction reaction is the rate-determining step affecting the OCV drop with respect to time. While the power density dropped to 158 mW·cm^−2^ for Nafion, it dropped more slowly to 180 mW·cm^−2^ for Nafion/CNCe, as shown in Figure 8b, because of the radical scavenging effect of CNCe during the hydrogen evolution reaction (HER) and oxygen reduction reaction (ORR).

## 4. Conclusions

The combined effect of the hybrid material CNCe is practically demonstrated for chemical stability and durability of composite membranes for the proton exchange membrane fuel cell application. HR-TEM images were recorded to identify the lattice sites of the cerium inside the crystalline structure of the CN. The presence of CNCe in Nafion was characterized by cross-sectional SEM images and EDX mapping. The oxidative stability and chemical structure degradation for the pristine and composite membrane were evaluated by Fenton’s test and in situ MEA degradation test. The results showed more stable chemical stability of composite membranes than pristine Nafion. The radical scavenging property of CNCe was further characterized by a polarization curve test for Nafion and Nafion/CNCe, showing excellent stability of Nafion/CNCe until 72 h under strong oxidative conditions. Overall, the combined effect of cerium into CN can be achieved, enhancing the durability of the membrane without sacrificing the migration of Ce from Nafion.

## Data Availability

Not applicable.

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
