# Peer review of "A Facile and Sustainable Enhancement of Anti-Oxidation Stability of Nafion Membrane"

_membranes, 2022, doi:10.3390/membranes12050521_

Round 1

Reviewer 1 Report

The manuscript reported the preparation of Ce grafted graphitic carbon nitrides (CNCe) nanoparticles, which are embedded in Nafion membranes to prolong the •OH radical scavenging effect with less reduction of proton conductivity, as the ionic interaction between Ce and π- electrons in g-C3N4 keeps Ce intact with sulfonic acid groups of Nafion molecules and prohibits its leaching out of the membrane. Compared with the pristine and Ce blended Nafion membranes, the CNCe embedded ones show tremendous improvement in long term anti-oxidation stability and single-cell performance.

I consider the content of this manuscript will definitely meet the reading interests of the readers of the Membranes journal. However, there are certain English spelling and grammar issues, and also the discussion and explanation should be further improved.

Therefore, I suggest giving a minor revision and the authors need to clarify some issues or supply some more experimental data to enrich the content. This could be a comprehensive and meaningful work after revision.

Detailed comments can be found in the attachment (PDF file).

Author Response

Reviewer 1:
The manuscript reported the preparation of Ce grafted graphitic carbon nitrides (CNCe) 
nanoparticles, which are embedded in Nafion membranes to prolong the •OH radical scavenging 
effect with less reduction of proton conductivity, as the ionic interaction between Ce and πelectrons in g-C3N4 keeps Ce intact with sulfonic acid groups of Nafion molecules and prohibits 
its leaching out of the membrane. Compared with the pristine and Ce blended Nafion membranes, 
the CNCe embedded ones show tremendous improvement in long term anti-oxidation stability 
and single-cell performance.
I consider the content of this manuscript will definitely meet the reading interests of the readers 
of the Membranes journal. However, there are certain English spelling and grammar issues, and 
also the discussion and explanation should be further improved.
Therefore, I suggest giving a minor revision and the authors need to clarify some issues or supply 
some more experimental data to enrich the content. This could be a comprehensive and 
meaningful work after revision.
Principal comments
1. Pay attention to grammar and spelling. The definite article is sometimes missing or redundant 
in some sentences throughout the manuscript, as well as the improper use of prepositions. Some 
other spelling typos are also found, I may point out some of them in the minor issues part (but 
impossible for everywhere). I suggest the authors check the above-mentioned issues throughout 
the whole manuscript.
Response: We appreciate the suggestion and the revised manuscript has been thoroughly 
checked with correction in the typo’s errors.
2. Line 102, ‘CNCe, 0.3 g, was subjected to sonication in DMSO for 60 min’. For solvent selection, 
why only DMSO is selected? Is there any special reason? Usually, the solvent selected for Nafion 
film casting is DMF or DMAC [Electrochimica Acta 49.19 (2004): 3211-3219]. And why the Nafion 
is cast by its original solvent, not drying Nafion first and then re-dissolve Nafion in DMF/DMSO? 
For example, ‘The solvent is removed from the 5 wt% Nafion solution by heating at 80 °C until a 
thin film is obtained. The film is then dissolved in DMF and mixed with the fillers disperion in DMF’ 
[Solid State Ionics 319 (2018): 110-116]. Normally, Nafion casted with DMF/DMSO demonstrate 
the highest mechanical strength [see Figure 13 Journal of Membrane Science 522 (2017): 56-67].
Response: Thank you for this comment. Yes, the solvents like DMF and DMAc provides very good 
mechanical stable film but our aim was to make a composite membrane with CNCe and we found 
extremely good dispersion of CNCe in DMSO. This was the reason for choosing DMSO instead of 
DMF and DMAc. 
3. Line 109, ‘The concentration of CNCe was kept at 6 wt% with respect to Nafion ionomer, 
because its concentration beyond 6 wt% causes decrement of proton conductivity...’ Although 
more than 6 wt% of filler, the ionic conductivity of hybrid membrane decreased significantly. 
However, the authors do not seem to be sure that 6wt% nano filler is the best load of hybrid 
membrane. In this case, it is necessary to study the hybrid membrane prepared by 2wt% and 4wt% 
Filler, physical and chemical properties and the properties of fuel cell single cell, so as to 
determine the optimal load of filler. The whole experiment has only one percentage of filler for 
the hybrid membrane, which can not form a complete research system. Therefore, I think it is 
necessary to supplement the hybrid membrane results of other filler loads.
Response: Thank you for the valuable suggestion. As we increased the content of cerium, we saw 
that it effectively decreased the proton conductivity value because of the neutralization of the 
sulfonic acid groups present on the sulfonated polymer. In previous reported literature we 
analyzed more than 8 wt % of cerium drastically decreases the proton conductivity. Therefore, in 
this study our aim was to load sufficient content of cerium to the membrane so that it can provide 
radical scavenging effect with stable proton conductivity. Further, for future work we are 
optimizing the filler content up to maximum values and we will definitely provide a 
comprehensive study for minimum to maximum content of filler with composite membrane for 
the convenience of the readers.
4. Line 169, for thermal stability (TGA), which atmosphere is used and how about the gas flow 
rate? Some more details should be provided.
Response: Thank you for this comment. TGA was performed under mixed gas environment i.e. 
N2 an air(Page-8, Paragraph-4)..
5. Line 281, for the TGA test, why only filler is tested, but not the hybrid membranes? I consider 
the analysis of thermal stability for Nafion and Nafion-hybrid membrane can make more sense, 
but it is still missing in the manuscript.
Response: We appreciate this comment. The main objective of this system was to enhance the 
antioxidation stability of the composite membranes for fuel cell application. Thermal study was 
conducted as a supplementary information to analyze the superior thermal behavior of the CNCe 
over Ce. In case of membranes the concentration was taken as fix i.e 6 wt% so we assumed that 
there would be not such a considerable change that can be found during TGA study. 
Secondary/Minor issues
For the Abstract, the length is a bit longer and appropriate reduction is required. The journal 
limits that ‘The abstract should be a total of about 200 words maximum’, see 
https://www.mdpi.com/journal/membranes/instructions.
For the Keywords, ‘cerium’, ‘Nafion’, and ‘proton conductivity’ should also be added to attract a 
broader readership.
Response: We altered the abstract as per the instructions and suggested keywords has been 
added to the revised manuscript. (Page-1, paragraph-1)
Line 8, ‘Although the cerium ion (Ce+3/+4, Ce) is reported as an effective •OH radical quencher.’ 
Ce should not be cerium ion, because it belongs to metal element. The content in brackets should 
be (Ce 3+/Ce 4+ ).
Response: It has been changed and added to revised manuscript.
Line 33, ‘The development of hydrocarbon-based membranes, however, is still being lagged 
because of its limited proton conductivity and high swelling...’ Also, the long-term chemical 
stability of hydrocarbon-based membranes such as SPEEK-based is not very good, even worse 
than that of Nafion membranes [Electrochimica Acta 309 (2019): 311-325]. This point should also 
be mentioned.
Response: This point has been added to the revised manuscript as per the suggestion. (Page-2, 
Paragraph-1)
Line 59, ‘(i) the mitigation of Ce ions from the membrane’. ‘form’ should be a spelling error.
Response: The error has been removed and corrected in the revised manuscript. (Page-3, 
paragraph-1)
Line 107, ‘The membranes were peeled off from the petri dish and then dipped into 1 M H2SO4 
solution for ionization of functional groups.’ For the pretreatment of Nafion, why is 3wt% H2O2 
solution not used? There should be some standard pretreatment procedures to follow [See 
Section 2.3, Solid State Ionics 319 (2018): 110-116].
Response: We thank you for the following comment. Yes, we are completely agreeing with the 
suggested procedure and the treatment of the synthesized membranes with 1M sulfuric acid was 
followed by several reported literatures.
Line 128, ‘The membrane samples were washed with DI water and then completely dried to 
measure their weights (in gram)’. How is the membrane dried? More details should be provided. 
For example, is it dried in airflow or heated in the oven at certain temperatures for certain hours?
Response: The following details has been added as per the suggestion. (Page- 8, paragraph-1).
Line 148, ‘Here is the hydroxide ion conductivity of the membrane in (S cm-1)’. No, it is wrong. 
Nafion is a proton exchange membrane, and it is pretreated in acid. Hence, the measured 
conductivity should be proton conductivity, not hydroxide ion conductivity.
Response: We sincerely apologies for the typing mistake. Now it has been changed in the revise 
manuscript. (Page-7, paragraph-3).
Equation 1 and Equation 3 should be consistent when expressing the dry membrane weight. Now 
in Equation 1, it is Wdry; in Equation 3, it is Dryw. It is so hard for the readers to understand in 
these complex descriptions.
Response: It has been changed in the revised manuscript as per the suggestion. (Page-7, 
paragraph-1).
Line 203, for the fuel cell test, how is the back pressure of the gases during the measurement? 
This information is still missing.
Response: The performance measurement for all synthetized membranes were conducted at a 
flow rate of 0.3 L min-1 with an operating pressure of approximate 30 psi.

Reviewer 2 Report

In their submission to Membranes entitled "A Facile and Sustainable Enhancement of Anti-oxidation Stability of Nafion Membrane", the authors, Sharma and Kim, report on the preparation In the study, the  Nafion/CNCe membranes displayed an enhanced long term anti-oxidation stability and high proton conductivity. The article is well written and conclusions are well suported by the experiments. The use of these materials seems to be promising My only comments are  (a) is that the authors should mention if conductivity studies were performed along or through the prepared membranes and (b) a comparison with other Nafion composite membarnes needs to be included to compare the performance (in terms of proton conductivity and fuel cell performance) of the reported membranes. Therefore, I recommend the publication of this manscript in Membranes.

Author Response

Reviewer 2:
In their submission to Membranes entitled "A Facile and Sustainable Enhancement of Antioxidation Stability of Nafion Membrane", the authors, Sharma and Kim, report on the preparation 
in the study, the Nafion/CNCe membranes displayed an enhanced long term anti-oxidation 
stability and high proton conductivity. The article is well written and conclusions are well 
suported by the experiments. The use of these materials seems to be promising My only 
comments are:
(a) is that the authors should mention if conductivity studies were performed along or through 
the prepared membranes,
Response: We thank you for this query and our response related to it has been added to the 
revised manuscript. (Page-7, Parahraph-2).
(b) A comparison with other Nafion composite membranes needs to be included to compare the 
performance (in terms of proton conductivity and fuel cell performance) of the reported 
membranes. Therefore, I recommend the publication of this manuscript in Membranes.
Response: Thank you for this comment. A table of comparison has been added in the revised 
manuscript (Page-19).
